# Hydrogen Peroxide Inhibits Hepatitis B Virus Replication by Downregulating HBx Levels via Siah-1-Mediated Proteasomal Degradation in Human Hepatoma Cells

**DOI:** 10.3390/ijms241713354

**Published:** 2023-08-28

**Authors:** Hyunyoung Yoon, Hye-Kyoung Lee, Kyung Lib Jang

**Affiliations:** 1Department of Integrated Biological Science, The Graduate School, Pusan National University, Busan 46241, Republic of Korea; yhypia@pusan.ac.kr (H.Y.); lhk6338@naver.com (H.-K.L.); 2Department of Microbiology, College of Natural Science, Pusan National University, Busan 46241, Republic of Korea; 3Microbiological Resource Research Institute, Pusan National University, Busan 46241, Republic of Korea

**Keywords:** HBx, hepatitis B virus, hydrogen peroxide, proteasome, Siah-1, p53

## Abstract

The hepatitis B virus (HBV) is constantly exposed to significant oxidative stress characterized by elevated levels of reactive oxygen species (ROS), such as H_2_O_2_, during infection in hepatocytes of patients. In this study, we demonstrated that H_2_O_2_ inhibits HBV replication in a p53-dependent fashion in human hepatoma cell lines expressing sodium taurocholate cotransporting polypeptide. Interestingly, H_2_O_2_ failed to inhibit the replication of an HBV X protein (HBx)-null HBV mutant, but this defect was successfully complemented by ectopic expression of HBx. Additionally, H_2_O_2_ upregulated p53 levels, leading to increased expression of seven in absentia homolog 1 (Siah-1) levels. Siah-1, an E3 ligase, induced the ubiquitination-dependent proteasomal degradation of HBx. The inhibitory effect of H_2_O_2_ was nearly abolished not only by treatment with a representative antioxidant, N-acetyl-L-cysteine but also by knockdown of either p53 or Siah-1 using specific short hairpin RNA, confirming the role of p53 and Siah-1 in the inhibition of HBV replication by H_2_O_2_. The present study provides insights into the mechanism that regulates HBV replication under conditions of oxidative stress in patients.

## 1. Introduction

The hepatitis B virus (HBV) is a significant human pathogen responsible for approximately 300 million chronic infections worldwide [1]. It is one of the leading causes of hepatic diseases such as acute and chronic hepatitis, cirrhosis, and hepatocellular carcinoma (HCC) [2,3]. As a member of the *Hepadnaviridae* family, HBV replicates and encapsidates a partially double-stranded circular DNA genome of around 3200 base pairs through reverse transcription of a pregenomic RNA (pgRNA) [4]. Among the four open reading frames (S, C, P, and X) in the HBV genome, the shortest X encodes a 17-kDa multifunctional protein known as HBV X protein (HBx). HBx, a potent HBV oncoprotein, localizes in the cytosol, nucleus, and mitochondria, affecting cell signaling, transcription, and mitochondrial function [2,5]. HBx also acts as a positive regulator of HBV replication, as demonstrated in various experimental systems, including human hepatocyte chimeric mice [6], infection cell culture [7], and mouse hydrodynamic injection models [8]. It stimulates the four viral promoters, resulting in the synthesis of HBV mRNA and pgRNA in the nucleus from a covalently closed circular DNA template [9,10]. Additionally, HBx indirectly contributes to HBV replication in the cytoplasm by deregulating cellular signal transduction such as the calcium signaling pathway [11] and the phosphatidylinositol 3-kinase/Akt pathway [12].

Reactive oxygen species (ROS), including hydrogen peroxide (H_2_O_2_), are highly reactive chemicals formed from oxygen (O_2_). Even in normal cells, ROS are continuously generated through processes such as oxidative phosphorylation in the mitochondria, protein folding in the endoplasmic reticulum, and the breakdown of lipids and amino acids in the cytosol [13]. Generally, ROS are considered toxic byproducts of cellular metabolism, exerting damaging effects, and are therefore subject to removal. However, ROS are also known to serve as signaling triggers, regulating diverse cellular processes such as proliferation, apoptosis, differentiation, and immune response against pathogens [13,14]. Among the various forms of ROS, H_2_O_2_ has been extensively studied as it plays a central role in redox signaling, regulation of oxidative stress, and biological activities [14,15,16]. Furthermore, H_2_O_2_ is known to cause damage to biomolecules such as proteins and DNA [17]. One representative factor that responds immediately to DNA damage caused by H_2_O_2_ is the tumor suppressor protein p53, which can induce growth arrest, senescence, and apoptosis, depending on the extent of the damage [18,19].

Previous reports have demonstrated that ROS levels are typically higher in the liver tissues and blood samples of patients infected with HBV [20,21]. Furthermore, several studies have shown that HBV infection in vitro can induce ROS production in cultured cells [22,23]. While other viral proteins including HBV surface antigen (HbsAg) and HBV core antigen (HbcAg) have been suggested to play roles [24], HBx appears to be the main contributor to the generation of oxidative stress during HBV infection. For example, HBx interacts with a voltage-dependent anion channel located in the mitochondria to alter its transmembrane potential [25]. Additionally, HBx lowers levels of mitochondrial enzymes involved in electron transport to elevate mitochondrial ROS levels [26]. Excess ROS produced during chronic HBV infection can damage cellular molecules like lipids, proteins, and DNA, which may contribute to the development of liver fibrosis and liver cancer [22,27]. Despite the established role of ROS during HBV pathogenesis [28], its effect on HBV replication remains obscure [23,29,30], primarily due to the lack of an effective in vitro HBV replication system so far.

Previous reports have demonstrated that HBx levels are known to be mostly regulated by an E3 ligase, seven in absentia homolog 1 (Siah-1), which can induce ubiquitin (Ub)-dependent proteasomal degradation of HBx [31,32]. In addition, p53 is known to activate Siah-1 expression through p53 response elements in the promoter [32]. Considering that p53 is a representative mediator of the DNA damage caused by excess ROS [18,19], it is intriguing to investigate whether ROS like H_2_O_2_ activates Siah-1 expression via activation of p53 and, thus, lowers HBx levels through Siah-1 mediated ubiquitination and proteasomal degradation to inhibit HBV replication. For this purpose, we employed here a recently optimized in vitro HBV replication system [33,34,35], which enables us to correctly evaluate the effect of ROS on HBV replication. Initially, it was examined whether H_2_O_2_ inhibits HBV replication in a p53-dependent fashion. Second, it was explored whether HBx is responsible for the p53-dependent inhibition of HBV replication by H_2_O_2_. Third, it was examined whether H_2_O_2_ activates Siah-1 expression through upregulation of p53 levels and subsequently downregulates HBx levels via Siah-1-mediated proteasomal degradation. Finally, we attempted to prove that H_2_O_2_ activates Siah-1 expression to downregulate HBx levels and inhibit HBV replication in an in vitro HBV replication system.

## 2. Results

### 2.1. H_2_O_2_ Inhibits HBV Replication in a p53-Dependent Fashion

Initially, we investigated whether H_2_O_2_ differently affects HBV replication in human hepatoma cell lines depending on the status of p53. HepG2 cells but not Hep3B cells express a wild type (WT) of functional p53 [36], providing a useful platform to study the roles of p53 in HBV-related molecular mechanisms [36]. HBV infections were performed on HepG2-NTCP and Hep3B-NTCP cells, which stably express the HBV receptor sodium-taurocholate co-transporting polypeptide (NTCP) [37], using HBV particles derived from a 1.2-mer HBV replicon, as previously described [38]. Replication of HBV in HepG2-NTCP and Hep3B-NTCP cells was evidenced by western blot analysis of viral proteins, such as HBx and HBsAg, in the total cell lysates (Figure 1a,e), and by measurement of virus particles in the culture supernatants by conventional PCR and quantitative real-time polymerase chain reaction (qPCR) (Figure 1b,f). Three different sizes of HBsAg, namely, large (L)-, middle (M)-, and small (S)-HBsAg, were found in the infected cells (Figure 1a,e), as previously demonstrated [38]. Unlike in HepG2-NTCP cells, small amounts of HBsAg derived from the integrated HBV genome [39] were detected in the cell lysates of uninfected Hep3B-NTCP cells (Figure 1e), although neither conventional PCR nor qPCR succeeded to detect evidence of HBV replication in these cells (Figure 1f). These results indicate that HBV successfully replicates in both HepG2-NTCP and Hep3B-NTCP cells under our experimental conditions.

Consistent with a previous report [40], HBV elevated the intracellular ROS levels during replication in both HepG2-NTCP and Hep3B-NTCP cells (Figure 1c,g). In addition, H_2_O_2_ treatment further increased ROS levels in the two cell lines in a dose-dependent fashion (Figure 1c,g). The effects of HBV infection and H_2_O_2_ treatment on intracellular ROS levels were higher in HepG2-NTCP cells than in Hep3B-NTCP cells (Figure 1c,g), indicating a possible role of p53 in the amplification of ROS levels during HBV infection and H_2_O_2_ treatment. Indeed, ectopic p53 expression not only elevated intracellular ROS levels, as previously demonstrated [41], but also allowed HBV infection and H_2_O_2_ treatment to elevate ROS levels in Hep3B-NTCP cells as efficiently as in HepG2-NTCP cells (Figure 1k). Taken together, we conclude that HBV infection and H_2_O_2_ treatment, individually or in combination, upregulate intracellular ROS levels, which can be further amplified by p53 in human hepatoma cells.

Consistent with a previous report demonstrating the role of p53 as a negative regulator of HBV propagation [38], the amounts of intracellular viral proteins and extracellular HBV particles were invariably higher in Hep3B-NTCP (Figure 1e,f) than in HepG2-NTCP cells (Figure 1a,b). In addition, ectopic p53 expression downregulated the levels of intracellular viral proteins and extracellular HBV particles in Hep3B-NTCP cells (Figure 1i,j). Moreover, H_2_O_2_ treatment strikingly downregulated the levels of intracellular HBV proteins and extracellular virions during HBV replication in HepG2-NTCP cells (Figure 1a,b), whereas these effects were negligible in Hep3B-NTCP cells (Figure 1e,f). Ectopic p53 expression was sufficient to enable H_2_O_2_ to inhibit HBV replication in Hep3B-NTCP cells (Figure 1i,j). The negative regulation of HBV replication by H_2_O_2_ appears to be a specific event since H_2_O_2_ at the concentration used in the present study did not exhibit noticeable side effects on the overall pattern of host cell protein synthesis nor the MTT activity as an indicator of cell viability, proliferation, and cytotoxicity (Figure 1a,d,e,h,i,l).

To confirm that H_2_O_2_ inhibits HBV replication in human hepatoma cells, we examined the effect of a representative antioxidant, N-acetyl-L-cysteine (NAC), on HBV replication in human hepatoma cells. Consistent with a previous report [23], NAC treatment effectively lowered ROS levels during HBV infection and, thus, almost abolished the ability of HBV to elevate ROS and p53 levels in HepG2-NTCP cells and Hep3B-NTCP cells expressing p53 (Figure 2a,b,d,e). In addition, NAC treatment activated HBV replication in HepG2-NTCP cells and Hep3B-NTCP cells expressing p53, as demonstrated by the step-wise increase in the levels of intracellular virus proteins and extracellular virus particles (Figure 2b,c,e,f). Taken together, we conclude that H_2_O_2_ inhibits HBV replication in human hepatoma cells in a p53-dependent fashion.

### 2.2. H_2_O_2_ Elevates p53 Levels to Inhibit HBV Replication in Human Hepatoma Cells

Next, we investigated how H_2_O_2_ inhibits HBV replication in a p53-dependent fashion. Consistent with previous reports [32,42], HBV upregulated p53 levels in HepG2-NTCP cells (Figure 1a). H_2_O_2_ treatment further elevated p53 levels in HepG2-NTCP cells in a dose-dependent fashion (Figure 1a). Accordingly, intracellular ROS and p53 levels were directly proportional to each other during HBV infection in HepG2-NTCP cells (Figure 1a,c). In contrast, H_2_O_2_ treatment lowered the levels of intracellular viral proteins and extracellular virions during HBV infection in HepG2-NTCP cells in a dose-dependent manner (Figure 1a,b), resulting in an inverse correlation between p53 and HBV levels under the condition (Figure 1a,b). Interestingly, H_2_O_2_ treatment also significantly increased intracellular ROS levels, whereas it hardly affected HBV propagation in Hep3B-NTCP cells, in which p53 was absent (Figure 1e–g). These results suggest that H_2_O_2_ inhibits HBV replication by elevating p53 levels in human hepatoma cells.

**Figure 1 ijms-24-13354-f001:**
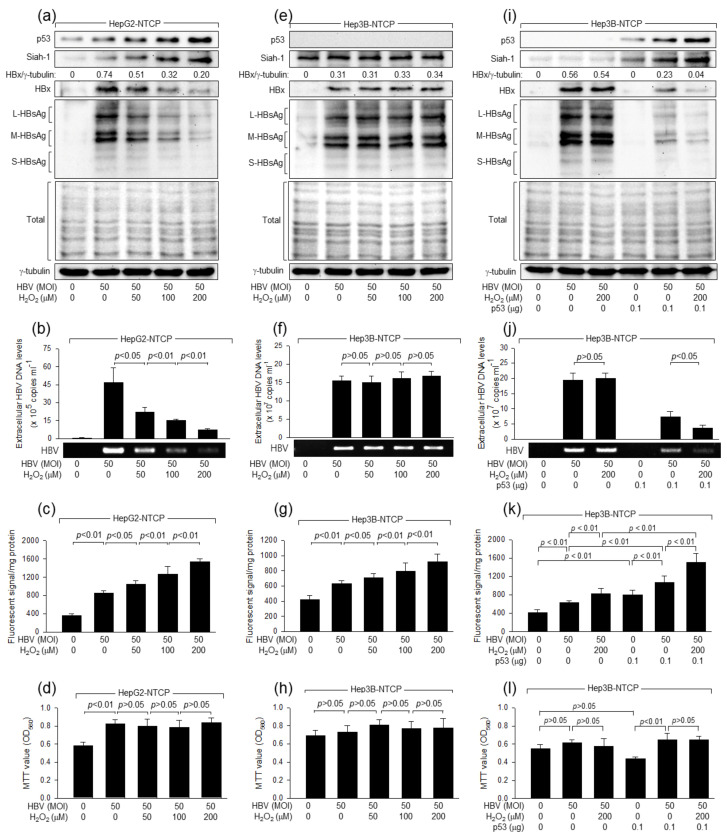
H_2_O_2_ inhibits HBV replication in a p53-dependent manner. HepG2-NTCP and Hep3B-NTCP cells were infected with HBV at the indicated multiplicity of infection (MOI) per cell for 24 h in DMEM containing 2% DMSO and 4% PEG 8000, washed twice with serum-free DMEM, and then incubated for an additional three days in DMEM containing 3% FBS, 2% DMSO, and 4% PEG 8000. Cells were either mock-treated or treated with the indicated concentrations of H_2_O_2_ for 24 h before harvesting. Cells were transfected with either an empty vector or p53 expression plasmid for 24 h prior to infection for (**i**–**l**). (**a**,**e**,**i**) Cell lysates were subjected to Western blotting to determine levels of p53, Siah-1, HBx, HBsAg, and γ-tubulin. The protein bands were quantified using Image J image analysis software version 1.8.0 (NIH) to show the level of HBx relative to the loading control (γ-tubulin). The SDS-PAGE gels stained with Coomassie brilliant blue were provided to show that H_2_O_2_ treatment did not non-specifically affect protein expression patterns under our experimental conditions. (**b**,**f**,**j**) The levels of HBV particles released from the cells prepared in (**a**,**e**,**i**) were measured by both conventional PCR and quantitative real-time PCR (qPCR). Results are shown as mean ± standard deviation obtained from four independent experiments (*n* = 4). (**c**,**g**,**k**) Levels of intracellular reactive oxygen species (ROS) were determined by using a fluorescent dye, chloromethyl dichlorodihydrofluorescein diacetate (CM-H_2_DCFDA), as described before [43]. (**d**,**h**,**l**) Cell viability was measured by MTT assays (*n* = 4).

**Figure 2 ijms-24-13354-f002:**
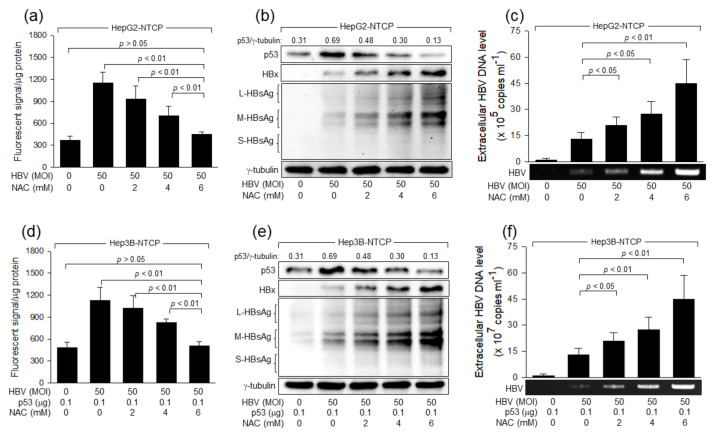
N-Acetylcysteine as an antioxidant stimulates HBV replication. HepG2-NTCP cells and Hep3B cells expressing p53 were infected with HBV, as described in Figure 1, for four days in the presence of the indicated concentration of NAC. (**a**,**d**) Intracellular ROS levels were determined as described in Figure 1c (*n* = 3). (**b**,**e**) Levels of the indicated proteins were determined by western blotting. (**c**,**f**) Levels of extracellular HBV DNA were determined by both conventional PCR and qPCR (*n* = 3).

To confirm that H_2_O_2_ upregulates p53 levels to inhibit HBV replication, we attempted to express ectopic p53 during HBV infection in Hep3B-NTCP cells. Indeed, H_2_O_2_ was able to inhibit HBV replication in Hep3B-NTCP cells with ectopic p53 expression, in which H_2_O_2_ upregulated p53 and ROS levels (Figure 1i–k). In addition, ectopic p53 expression without H_2_O_2_ treatment could inhibit HBV replication in Hep3B-NTCP cells, in which p53 elevated intracellular ROS levels (Figure 1i–k). Taken together, we conclude that H_2_O_2_ upregulates p53 levels to inhibit HBV replication in human hepatoma cells.

### 2.3. H_2_O_2_ Downregulates HBx Levels to Inhibit HBV Replication in Human Hepatoma Cells

The detailed mechanism by which H_2_O_2_ inhibits HBV replication in a p53-dependent fashion was investigated. According to data obtained with an in vitro HBV infection system, H_2_O_2_ dose-dependently lowered the levels of HBx, a positive regulator of HBV replication [6,7,8,9,10], in HepG2-NTCP cells but not in Hep3B-NTCP cells (Figure 1a,e). In addition, ectopic p53 expression enabled H_2_O_2_ to downregulate HBx levels in Hep3B-NTCP cells (Figure 1i). These results suggest that HBx mediates the potential of H_2_O_2_ to inhibit HBV replication in a p53-dependent mechanism. To prove that H_2_O_2_ inhibits HBV replication by lowering HBx levels, a 1.2-mer WT HBV replicon (1.2-mer WT) and its HBx-null counterpart (1.2-mer HBx-null) [44] were employed. Transient transfection of HepG2 cells with 1.2-mer WT resulted in the generation of ROS (Figure 3a), upregulation of p53 levels (Figure 3b), production of intracellular HBV proteins, including HBx and HBsAg (Figure 3b), and secretion of extracellular virus particles (Figure 3c). Consistent with data obtained with an in vitro HBV infection system (Figure 1a,c), H_2_O_2_ treatment further elevated intracellular ROS and p53 levels in HepG2 cells transfected with 1.2-mer WT in a dose-dependent fashion (Figure 3a,b). In addition, H_2_O_2_ treatment inhibited replication of HBV derived from 1.2-mer WT in HepG2 cells, as evidenced by the lowered levels of intracellular HBV proteins and extracellular virus particles (Figure 3b,c). These results indicate that the negative regulation of HBV replication by H_2_O_2_ can be exactly reproduced in a 1.2-mer HBV replicon system.

Transfection of 1.2-mer HBx-null also significantly increased ROS levels but failed to upregulate p53 levels in HepG2 cells, presumably because the ROS levels induced by HBV infection in the absence of HBx were too low to upregulate p53 levels in these cells (Figure 3d,e). In addition, transfection of 1.2-mer HBx-null yielded lower levels of intracellular HBV proteins and extracellular virus particles (Figure 3e,f), compared to those obtained with 1.2-mer WT (Figure 3b,c), as previously demonstrated [38,44]. The defects of 1.2-mer HBx-null in ROS generation, p53 activation, and HBV replication were almost completely complemented by ectopic HBx expression (Figure 3d–f, lane 6). Moreover, ectopic HBx expression alone was sufficient to upregulate ROS and p53 levels in HepG2 cells (Figure 3d,e, lane 5). These results strongly argue that HBx plays essential roles in ROS generation, p53 activation, and virus replication during HBV infection in human hepatoma cells, which is consistent with previous reports [25,26,32,38]. Interestingly, although H_2_O_2_ treatment normally elevated intracellular ROS and p53 levels in HepG2 cells transfected with 1.2-mer HBx-null, it failed to inhibit HBV replication in these cells (Figure 3d–f, lane 4). In addition, ectopic HBx expression allowed H_2_O_2_ to inhibit the propagation of HBV derived from 1.2-mer HBx-null in HepG2 cells (Figure 3e,f, lane 8). Taken together, we conclude that H_2_O_2_ lowered HBx levels to inhibit HBV replication in human hepatoma cells.

### 2.4. H_2_O_2_ Upregulates p53 Levels to Downregulate HBx Levels in Human Hepatoma Cells

We investigated how H_2_O_2_ downregulates HBx levels in human hepatoma cells. According to data from both HBV infection and HBx overexpression systems, H_2_O_2_ invariably downregulated HBx levels, whereas it upregulated p53 levels in human hepatoma cells (Figure 1a and Figure 3e). The inverse correlation between p53 and HBx during HBV infection was confirmed by flow cytometric analysis (Figure 4a). HBV infection at a dose of 50 genome equivalents (GEQ) per cell for 4 days resulted in HBV replication in a majority of the cells, as demonstrated by 70.4% of HBx positivity. Consistent with data from the Western blot analysis (Figure 1a), both H_2_O_2_ and HBV individually increased the percentages of HepG2-NTCP cells with the indicated fluorescent signal of p53. Interestingly, however, the ability of H_2_O_2_ to increase the p53 signal was lower in the HBV-infected HepG2-NTCP cells (51.0%) than in the mock-infected ones (81.2%), indicating that HBx weakens the ability of H_2_O_2_ to upregulate p53 levels in human hepatoma cells. H_2_O_2_ treatment decreased the HBx signal, whereas it augmented the p53 signal in HepG2-NTCP cells infected with HBV, resulting in an inverse correlation between p53 and HBx levels. Unlike in HepG2-NTCP cells, the effect of H_2_O_2_ on the HBx signal was not observed during HBV replication in Hep3B-NTCP cells, in which any p53 signal was not detected. Taken together, it was possible to hypothesize that H_2_O_2_ upregulates p53 levels to downregulate HBx levels during HBV replication in human hepatoma cells.

Downregulation of HBx levels can be either a cause or a result of the H_2_O_2_-induced inhibition of HBV replication. This question can be answered by verifying whether H_2_O_2_ can downregulate ectopic HBx levels in the absence of HBV replication. Indeed, H_2_O_2_ downregulated ectopically expressed HBx levels in HepG2 cells, in which H_2_O_2_ upregulated p53 levels (Figure 4b). In contrast, H_2_O_2_ treatment failed to induce any detectable changes in HBx levels in Hep3B cells, in which p53 was absent (Figure 4b). To confirm the role of p53 in the H_2_O_2_-induced downregulation of HBx levels in human hepatoma cells, we attempted to knock down p53 in HepG2 cells and to ectopically express p53 in Hep3B cells before H_2_O_2_ treatment. As a result, p53 knockdown upregulated HBx levels in HepG2 cells treated with H_2_O_2_ (Figure 4c), whereas ectopic p53 expression enabled H_2_O_2_ to downregulate HBx levels in Hep3B cells, in which H_2_O_2_ upregulated p53 levels in a dose-dependent fashion (Figure 4d). Moreover, overexpression of p53 without H_2_O_2_ treatment was sufficient to downregulate HBx levels in both HepG2 and Hep3B cells (Figure 4e,f). Taken together, we conclude that H_2_O_2_ elevated p53 levels to lower HBx levels during HBV replication in human hepatoma cells.

### 2.5. H_2_O_2_ Upregulates Siah-1 Levels through Activation of p53 to Downregulate HBx Levels

Previous reports have demonstrated that p53 transcriptionally activates the Siah-1 gene, whose product acts as an E3 ligase of HBx [31,45]. It has been further demonstrated that HBx-mediated activation of p53 leads to upregulation of Siah-1 levels, which in turn downregulates HBx levels via Ub-dependent proteasomal degradation to complete a negative feedback loop [32]. Consistently, HBV infection upregulated Siah-1 levels in human hepatoma cells only in the presence of p53 (Figure 1a,e). In addition, transient transfection of 1.2-mer WT but not 1.2-mer HBx-null upregulated Siah-1 levels in HepG2 cells (Figure 3b,e). Moreover, ectopic HBx expression without the involvement of other viral proteins was sufficient to elevate Siah-1 levels in HepG2 cells, whereas this effect was not observed in Hep3B cells (Figure 4b). H_2_O_2_ treatment in the presence of p53 further elevated Siah-1 levels in the HBV-infected cells (Figure 1a) as well as in the HBx-expressing cells (Figure 4b). Therefore, it was possible to hypothesize that H_2_O_2_ and HBx, individually or in combination, activate Siah-1 expression in human hepatoma cells via activation of p53. The role of p53 in the activation of Siah-1 expression by HBx and H_2_O_2_ in human hepatoma cells was confirmed by both p53 knockdown and p53 overexpression, which resulted in a decrease in Siah-1 levels in HepG2 cells and an increase in Siah-1 levels in Hep3B cells, respectively (Figure 4c,d). Moreover, ectopic p53 expression without H_2_O_2_ treatment and HBx expression was sufficient to elevate Siah-1 levels in both HepG2 and Hep3B cells (Figure 4e,f).

Having established that H_2_O_2_ upregulates Siah-1 levels in HBx-expressing cells, it was investigated whether Siah-1 is actually implicated in the p53-dependent downregulation of HBx levels by H_2_O_2_ in human hepatoma cells. H_2_O_2_ treatment downregulated HBx levels in HepG2 cells as it upregulated Siah-1 levels through activation of p53 in HepG2 cells (Figure 4b). In contrast, neither Siah-1 upregulation nor HBx downregulation was observed in Hep3B cells upon H_2_O_2_ treatment (Figure 4b). In addition, ectopic Siah-1 expression lowered HBx levels, whereas Siah-1 knockdown elevated HBx levels in human hepatoma cells, regardless of the presence of p53 and H_2_O_2_ (Figure 5a,b), confirming the p53-independent action of Siah-1 on HBx [46]. Ectopic Siah-1 expression downregulated p53 levels in HepG2 cells, presumably through downregulation of HBx levels, whereas Siah-1 knockdown upregulated p53 levels in Hep3B cells, possibly through upregulation of HBx levels (Figure 5a,b). Taken together, we concluded that H_2_O_2_ upregulates Siah-1 level through activation of p53 to downregulate HBx levels in human hepatoma cells.

### 2.6. H_2_O_2_ Induces Siah-1-Mediated Proteasomal Degradation of HBx in a p53-Dependent Fashion

Having established that H_2_O_2_ lowers HBx levels through p53-dependent upregulation of Siah-1 levels, we examined whether H_2_O_2_ actually decreases the protein stability of HBx in a p53- and Siah-1-dependent mechanism. For this purpose, we treated HBx-expressing cells with cycloheximide (CHX) to block further protein synthesis and measured HBx and γ-tubulin levels in these cells (Figure 5c). The normal half-life (t_1/2_) of HBx was 57.7 min in HepG2 cells, which was shortened by H_2_O_2_ treatment to 22.3 min. However, the t_1/2_ value of HBx in Hep3B cells was 115.5 min, which was minimally affected by H_2_O_2_ treatment (t_1/2_ = 115.5 min). These results suggest that H_2_O_2_ decreases the protein stability of HBx in a p53-dependent mechanism. The knockdown of Siah-1 abolished the potential of H_2_O_2_ to decrease HBx stability, and thus fully restored the t_1/2_ value of HBx in HepG2 cells in the presence of H_2_O_2_ to 53.3 min, indicating that H_2_O_2_ largely downregulates HBx levels in HepG2 cells via upregulation of Siah-1 levels. The knockdown of Siah-1 in Hep3B cells also lengthened the t_1/2_ value of HBx in Hep3B cells to 173.2 min, confirming that Siah-1 acts on HBx in a p53-independent mechanism.

To further demonstrate the role of Siah-1 in H_2_O_2_-induced downregulation of HBx levels in the presence of p53, we examined whether H_2_O_2_ would increase the Siah-1-mediated ubiquitination of HBx in a p53-dependent fashion. For this purpose, we introduced HBx and HA-tagged Ub into HepG2 and Hep3B cells with or without H_2_O_2_ treatment and immunoprecipitated the Ub-complexed HBx. According to data from the co-immunoprecipitation (co-IP), Siah-1 interacted with HBx to induce its ubiquitination, as demonstrated by the smeared multiple bands of Ub(n)-HBx in HepG2 and Hep3B cells (Figure 5d, lanes 2 and 6). H_2_O_2_ treatment strengthened the interaction between HBx and Siah-1 in HepG2 cells, resulting in strong ubiquitination of HBx and the subsequent downregulation of its protein levels, whereas these effects were largely negligible in Hep3B cells (Figure 5d, lanes 3 and 7). In addition, the knockdown of Siah-1 in the presence of H_2_O_2_ weakened the interaction between HBx and Siah-1 in HepG2 and Hep3B cells, resulting in a decrease in the ubiquitination of HBx and the subsequent elevation of its protein levels (Figure 5d, lanes 4 and 8). Consistently, treatment with a universal proteasomal inhibitor MG132 nearly abolished the ability of H_2_O_2_ to alter p53, Siah-1, and HBx levels and equalized the protein levels of HBx regardless of the presence of H_2_O_2_ (Figure 5e), confirming that H_2_O_2_ lowers HBx levels through Siah-1-mediated ubiquitination and proteasomal degradation.

### 2.7. H_2_O_2_ Induces Siah-1-Mediated Proteasomal Degradation of HBx to Inhibit HBV Replication in Human Hepatoma Cells

Previous reports have shown that HBx enhanced HBV replication through activation of the core promoter, increasing transcription of pgRNA from cccDNA [9,10]. Consistently, HBx significantly increased the luciferase activity from pHBV-luc containing the full-length HBV core promoter in HepG2 and Hep3B cells (Figure 6a,b). Treatment with H_2_O_2_ reduced the HBx-mediated activation of the HBV core promoter in HepG2 cells (Figure 6a), but not in Hep3B cells (Figure 6b). Therefore, we conclude that H_2_O_2_ inhibits the HBV core promoter via the p53-dependent downregulation of HBx levels, whereas it hardly affects the p53-independent activation of the HBV core promoter by HBx.

Consistent with data obtained from the HBx overexpression system (Figure 4 and Figure 5), H_2_O_2_ treatment upregulated both p53 and Siah-1 levels during HBV replication in HepG2-NTCP cells, resulting in inhibition of HBV replication, as evidenced by the lower levels of intracellular HBV proteins, including HBx and HBsAg (Figure 6c), and extracellular HBV particles (Figure 6e), whereas none of these effects were observed in Hep3B-NTCP cells (Figure 6d,f). In addition, Siah-1 knockdown nearly abolished the potential of H_2_O_2_ to inhibit HBV replication in HepG2-NTCP cells, as demonstrated by the full recovery in the levels of extracellular HBV particles as well as intracellular HBV proteins (Figure 6c,e). Siah-1 knockdown also stimulated HBV replication in Hep3B-NTCP cells (Figure 6d,f), which is consistent with the p53-independent degradation of HBx by Siah-1 (Figure 5c,d). Taken together, we concluded that H_2_O_2_ inhibits HBV replication by stimulating Siah-1-mediated ubiquitination and proteasomal degradation of HBx in a p53-dependent fashion.

## 3. Discussion

HBV infection generally induces oxidative stress, which is usually characterized by elevated levels of ROS such as H_2_O_2_ in the liver and blood of patients [20,21]. Both host and viral factors are implicated in this process [20,21]. It is relatively well established that the virus-specific cytotoxic T lymphocytes (CTLs), which are primarily responsible for liver injury by eliminating infected hepatocytes and inducing the production of inflammatory cytokines [47], play a major role in the generation of ROS in the liver. HBV itself is also known to contribute to the accumulation of ROS in the hepatocytes of the liver, which is primarily mediated by HBx [48,49,50]. Consistently, the present study showed that HBV infection resulted in elevated ROS levels in cultured human hepatoma cells (Figure 1c). The role of HBx in this process was proven by ectopic HBx expression, which was sufficient to elevate ROS levels without the involvement of other viral proteins (Figure 3d). However, neither HBV infection nor HBx expression was able to elevate ROS levels above twofold (Figure 1c and Figure 3d), as previously demonstrated [51], primarily due to a lack of CTL responses during HBV infection in cultured cells. Therefore, we employed H_2_O_2_ in this study to induce adequate intracellular ROS levels and to properly evaluate the effect of H_2_O_2_ on HBV replication in cultured cells.

Only a few studies have been reported on the relation between ROS and HBV replication mainly because an effective in vitro HBV replication system was not available until recently. Early studies demonstrated that H_2_O_2_ not only decreases the synthesis of viral proteins, including HBsAg, HBeAg, and other viral proteins in the infected cells but also reduces the release of progeny HBV particles from the infected cells [29,30]. It was also demonstrated that the antioxidant NAC almost abolished the antiviral effect of H_2_O_2_, confirming the role of H_2_O_2_ as an inhibitor of HBV replication [29]. A completely opposite role of H_2_O_2_ in the regulation of HBV replication has also been reported. According to a more recent report by Ren et al. [23], H_2_O_2_ promotes HBV replication in cell culture systems, whereas removal of ROS by treatment with NAC results in inhibition of HBV replication. As more related data is not available, it is not easy to reconcile the discrepancy on this issue. Differences in experimental conditions such as HBV genotype, host cell, and HBV infection system can be considered. In fact, the previous studies employed different HBV-producing cell lines established by transfection of different HBV DNA clones into three human hepatoma cell lines, Huh-7, HB611, and HepG2, in which H_2_O_2_ may differently affect HBV replication [23,29,30]. In addition, considering the limited HBV reinfection rate mainly due to defects of the HBV receptor NTCP in human hepatoma cell lines [37], the HBV infection models employed by the previous reports may not properly mimic the natural course of HBV infection in patients. The inefficient HBV replication may also limit statistical analysis of the data obtained from the HBV replication systems. Recently, an optimized HBV infection system, which was established by expressing NTCP in human hepatoma cells and including dimethyl sulfoxide (DMSO) and polyethylene glycol 8000 (PEG 8000) in culture media, allowed robust HBV infection with low starting inoculum concentrations (10 to 100 GEQ per cell) and short incubation periods (4 to 7 days) [33,34,35]. Consistently, infection of HepG2-NTCP cells with HBV at a multiplicity of infection (MOI) of 50 for 4 days in the presence of DMSO and PEG 8000 enabled us to quantitatively analyze the HBV replication rates via measurement of HBV proteins in the cell lysates and HBV particles in the culture media (Figure 1). The present study provides two critical pieces of evidence supporting that H_2_O_2_ inhibits HBV replication in human hepatoma cells. First, H_2_O_2_ treatment not only downregulated intracellular HBV proteins such as HBx and HBsAg during HBV infection but also decreased the release of extracellular HBV particles from the infected cells (Figure 1). Second, treatment with the antioxidant NAC stimulated HBV replication, as demonstrated by an increase in levels of intracellular HBV proteins in the infected cells and extracellular HBV particles released from the infected cells (Figure 2). While the present study highlights NAC as a representative antioxidant that inhibits HBV replication, it is important to consider that NAC’s actions may not be solely limited to its antioxidant properties. NAC, aside from being a precursor for the antioxidant glutathione, is recognized for its interactions with various components of cell signaling pathways [52]. In addition to ROS, a range of stress responses like ER stress, heat shock response, DNA damage response, apoptosis, and autophagy can all have an influence on virus replication levels. In the context of the present study, the negative effect of H_2_O_2_ on HBV replication might not result from its cytotoxicity, as demonstrated by the MTT assay data (Figure 1d).

H_2_O_2_ induces single- and double-strand DNA breaks, resulting in activation of DNA damage signaling pathways mediated by ATR/Chk1 and ATM/Chk2, respectively, and subsequent upregulation of p53 levels [17]. In this study, two human liver cancer cell lines, HepG2 and Hep3B, were employed to investigate the potential role of p53 in the regulation of HBV replication by H_2_O_2_, because HepG2 cells, but not Hep3B cells, express a functional form of p53, while sharing several well-characterized characteristics [36,53]. H_2_O_2_ inhibited HBV replication in HepG2-NTCP cells, in which H_2_O_2_ upregulated p53 levels (Figure 1a,b), but not in Hep3B-NTCP cells (Figure 1e,f), suggesting a possible role of p53 in the regulation of HBV replication. However, the two cell lines exhibit several other differences, such as ethnic origins, distinct chromosome abnormalities, HBV DNA integration, and tumorigenicity [36,53], which may affect the effect of H_2_O_2_ in these two cell lines. This question can be answered by examining the effect of H_2_O_2_ in Hep3B cells after p53 complementation and in HepG2 cells after p53 knockdown. Indeed, ectopic p53 expression successfully restored the potential of H_2_O_2_ to inhibit HBV replication in Hep3B-NTCP cells, in which H_2_O_2_ upregulated exogenous p53 levels (Figure 1i,j). Moreover, p53 knockdown almost abolished the potential of H_2_O_2_ to downregulate HBx levels in HepG2 cells (Figure 4c). It is obvious, therefore, that p53 upregulation is responsible for the H_2_O_2_-induced inhibition of HBV replication, which is consistent with the role of p53 as a negative regulator of HBV replication [38].

It has been demonstrated that p53 downregulates HBx levels via Ub-dependent proteasomal degradation, resulting in inhibition of HBV replication [32,54]. The present study provides several lines of evidence supporting that H_2_O_2_ inhibits HBV replication by lowering HBx levels in a p53-dependent fashion. Firstly, H_2_O_2_ failed to inhibit replication of the HBV mutant derived from 1.2-mer HBx-null in HepG2 cells, in which H_2_O_2_ upregulated p53 levels (Figure 3e,f). Secondly, ectopic HBx expression was sufficient for H_2_O_2_ to inhibit the propagation of HBx-null HBV in HepG2 cells (Figure 3e,f). Thirdly, H_2_O_2_ downregulates HBx levels via Ub-dependent proteasomal degradation in human hepatoma cells in a p53-dependent fashion (Figure 4 and Figure 5). Considering the potential of HBx to stimulate HBV replication [9,10], which was also demonstrated in the present study (Figure 3e,f), the p53-dependent downregulation of the HBx levels may play a critical role in the inhibition of HBV replication by H_2_O_2_.

H_2_O_2_ treatment invariably elevated intracellular ROS to higher levels in HepG2-NTCP than in Hep3B-NTCP cells (Figure 1c,g), obviously due to the p53-mediated amplification of ROS in HepG2-NTCP cells [55]. Thus, it is possible to consider that p53 contributes to the H_2_O_2_-induced inhibition of HBV replication simply by elevating intracellular ROS levels in HepG2-NTCP cells. Interestingly, however, although H_2_O_2_ treatment of HepG2-NTCP cells at 50 μM and Hep3B-NTCP cells at 200 μM resulted in similar levels of intracellular ROS, the negative effect of H_2_O_2_ on HBV replication was observed only in HepG2-NTCP cells (Figure 1). Therefore, more complicated roles of p53 appeared to be involved in the H_2_O_2_-induced inhibition of HBV replication. Previous reports demonstrated that Siah-1 as an E3 ligase of HBx mediates the potential of p53 to downregulate HBx levels [31,32]. Consistently, the present study showed that H_2_O_2_ upregulated Siah-1 levels in both HBx overexpression and HBV replication systems via upregulation of p53 (Figure 1, Figure 3 and Figure 4). Additionally, the present study provided several lines of evidence supporting that H_2_O_2_ inhibits HBV replication by lowering HBx levels via Siah-1-mediated proteasomal degradation. Firstly, H_2_O_2_ induced Siah-1-mediated polyubiquitination proteasomal degradation of HBx in a p53-dependent fashion (Figure 5). Secondly, H_2_O_2_ inhibited HBV replication by elevating Siah-1 levels and lowering HBx levels (Figure 6). Thirdly, all the effects of H_2_O_2_ became invalid when Siah-1 was knocked down (Figure 5 and Figure 6), providing direct evidence for the role of Siah-1 in the H_2_O_2_-induced downregulation of HBx levels and subsequent inhibition of HBV replication.

Like H_2_O_2_ treatment, HBx itself is also known to upregulate intracellular ROS and p53 levels [32,48,49,50], which was also demonstrated in the present study (Figure 3). HBx may contribute to the H_2_O_2_-induced inhibition of HBV replication by lowering HBx levels via a negative feedback loop. It is unknown why HBx restricts its own level during HBV replication via a negative feedback loop involving ROS and p53. HBx may control HBV replication within a certain range by modulating its own protein level during a long course of persistent infection. Otherwise, the negative regulation of HBV replication by H_2_O_2_ may serve as an innate host defense system against HBV infection, which adds an additional role of p53 as the guardian of the genome. Further studies are needed to clarify this issue. Human primary hepatocytes and humanized mice may be employed to more precisely evaluate the effect of H_2_O_2_ on HBV replication. In conclusion, the present study provides insights into the mechanism underlying the regulation of HBV replication under oxidative stresses in patients.

## 4. Materials and Methods

### 4.1. Plasmid

The plasmid pCMV-3 × HA1-HBx (HA-HBx) encodes full-length HBx (genotype D) downstream of the three copies of the influenza virus haemagglutinin (HA) [56]. The 1.2-mer WT HBV replicon containing 1.2 units of the HBV genome (genotype D) and its HBx-null counterpart [44] were kindly provided by W. S. Ryu (Yonsei University, Seoul, Republic of Korea). The HBV core promoter/enhancer reporter construct, pHBV-luc, was previously described [44]. The plasmid RC210241 (Cat No. 003049), encoding the human NTCP, was obtained from OriGene (Rockville, MD, USA). The plasmid pSiah-1-Myc WT, encoding Myc-tagged Siah-1, has been previously described [45]. Scrambled (SC) short hairpin RNA (shRNA) (Cat No. sc-37007) and p53 shRNA (Cat No. sc-29435) were purchased from Santa Cruz Biotechnology (Santa Cruz, CA, USA). Siah-1 shRNA (Cat No. SHCLND-NM003031) was obtained from Sigma-Aldrich (St. Louis, MO, USA). Plasmid pCH110 (Cat No. 27-4508-01), encoding the *Escherichia coli* β-galactosidase (β-Gal) gene, was obtained from Addgene (Watertown, MA, USA). The pCMV p53 encoding WT p53 was a gift from Chang-Woo Lee (Sungkyunkwan University, Suwon, Republic of Korea). The pHA-Ub encoding HA-tagged Ub was kindly provided by Y. Xiong (University of North Carolina at Chapel Hill, Chapel Hill, NC, USA).

### 4.2. Cell Culture and Transfection

The cell lines HepG2 (Cat No. 88065) and Hep3B (Cat No. 88064) were obtained from the Korean Cell Line Bank (KCLB, Seoul, Republic of Korea). Stable cell lines, HepG2-NTCP and Hep3B-NTCP, were established by transfection with RC210241, followed by selection with 500 μg·mL^−1^ G418 sulfate (Sigma-Aldrich, Cat No. A1720). Cells were cultured in Dulbecco Modified Eagle Medium (DMEM) (WelGENE, Gyeongsan, Republic of Korea, Cat No. LM001-05) supplemented with 10% fetal bovine serum (FBS, Capricorn Scientific, Ebsdorfergrund, Germany, Cat No. FBS-22A), 100 units·mL^−1^ penicillin G (Sigma-Aldrich, Cat No. P3032), and 100 μg·mL^−1^ streptomycin (United States Biological, Salem, MA, USA, Cat No. 21865) at 37 °C in 5% CO_2_-humidified atmosphere. For transient expression, 2 × 10^5^ cells per well in a 6-well plate were transfected using TurboFect transfection reagent (Thermo Fisher Scientific, Waltham, MA, USA, Cat No. R0532) according to the manufacturer’s instructions with the designated amounts of plasmids and an empty vector supplemented to make the final amount of cocktails equivalent. The cells were treated with H_2_O_2_ (Sigma-Aldrich, Cat No. H1009), CHX (Sigma-Aldrich, Cat No. C7698), NAC (Sigma-Aldrich, Cat No. A7250), or MG132 (Millipore, Burlington, MA, USA, Cat No. 474790), under the indicated conditions.

### 4.3. HBV Cell Culture System

For HBV stock preparation, Hep3B-NTCP cells were transiently or stably transfected with the 1.2-mer HBV replicon plasmid as described above. The culture supernatant was collected for the preparation of HBV stocks. HBV titers were determined by qPCR, as described in the next section. HBV infection was conducted in 6-well plates at an MOI of 50 for 4 days, according to an optimized HBV cell culture system with slight modifications [33,34]. Briefly, 2 × 10^5^ cells were inoculated with 1 × 10^7^ GEQ of HBV and incubated for 24 h in serum-free DMEM containing 4% PEG 8000 (Sigma-Aldrich, Cat No. D4463) and 2% DMSO (Sigma-Aldrich, Cat No. D8418). After washing twice with PBS, the cells were incubated in DMEM supplemented with 3% FBS, 4% PEG 8000, and 2% DMSO for an additional three days.

### 4.4. Quantitative Real-Time PCR of HBV DNA

The extracellular HBV titers were measured by qPCR, as described previously [38]. Briefly, HBV genomic DNA was purified from the culture supernatant using the QIAamp DNA Mini Kit (Qiagen, Hilden, Germany, Cat No. 51306). For conventional PCR analysis of HBV DNA, genomic DNA was amplified using 2 × Taq PCR Master Mix 1 (BioFACT, Daejeon, Republic of Korea, Cat No. ST301-19h) and a primer pair, HBV 1399F (5′-TGG TAC CTG CGC GGG ACG TCC TT-3′) and HBV 1632R (5′-AGC TAG CGT TCA CGG TGG TCT CC-3′). For qPCR analysis, HBV DNA was amplified using the SYBR premix Ex Taq II (Takara Bio, Kusatsu, Japan, Cat No. RR82LR) and HBV 379F (5′-GTG TCT GCG GCG TTT TAT CA-3′) and HBV 476R (5′-GAC AAA CGG GCA ACA TAC CTT-3′) using a Rotor-gene qPCR machine (Qiagen).

### 4.5. Determination of Intracellular ROS Levels

Intracellular ROS levels were measured using chloromethyl dichlorodihydrofluorescein diacetate (CM-H_2_DCFDA; Invitrogen, Waltham, MA, USA, Cat No. C6827), which is widely applied as an H_2_O_2_-specific probe in intact cells [43]. Briefly, 1 × 10^5^ cells per well in 12-well plates were treated with 10 µM CM-H_2_DCFDA in serum-free media for 30 min. After washing with PBS, cells were collected by treatment with Trypsin-EDTA (Gibco, Grand Island, NY, USA, Cat No. 25200-072). The oxidation of CM-H_2_DCFDA to a green fluorescent product, DCF, was then quantified using a microplate reader (Mithras LB940, Berthold Technologies, Bad Wildbad, Germany) at excitation and emission wavelengths of 485 nm and 535 nm, respectively.

### 4.6. Luciferase Reporter Assay

Approximately 1 × 10^5^ cells per well in 12-well plates were transfected with 0.3 µg of pHBV-luc along with the indicated plasmids under the indicated conditions. To control for transfection efficiency, 0.1 µg pCH110 was co-transfected as an internal control. At 48 h after transfection, a luciferase assay was performed using the Luciferase Reporter 1000 Assay System (Promega, Madison, WI, USA, Cat No. E4550). Luciferase activity was measured using a microplate luminometer (LuBi, MicroDigital, Seongnam, Republic of Korea). β-gal activity was measured using a β-gal assay kit (Thermo Fisher Scientific, Cat No. 34055). Luciferase activity was normalized to the β-gal activity measured in the corresponding cell extracts.

### 4.7. Western Blot Analysis

The plasmid Cells were lysed in buffer (50 mM Tris-HCl, pH 8.0, 150 mM NaCl, 0.1% SDS, and 1% NP-40) supplemented with protease inhibitors (Roche, Basel, Switzerland, Cat No. 11836153001). Protein concentrations of cell extracts were measured using a protein assay kit (Bio-Rad, Hercules, CA, USA, Cat No. 5000006). Cell extracts were separated by SDS-PAGE and the gel was stained with Coomassie R250 staining solution (Bio-Rad, Cat No. 161-0400) for 30 min and then destained using a destaining solution composed of 40% methanol and 10% acetic acid. Proteins transferred onto a nitrocellulose blotting membrane (Amersham, UK, Cat No. 10600003) were incubated with primary antibodies against Siah-1 (Abcam, Cambridge, UK, Cat No. ab2237, 1:2000 dilution); p53 (Santa Cruz Biotechnology, Cat No. sc-126, 1:1000 dilution); HBsAg (Santa Cruz Biotechnology, Cat No. sc-53300, 1:400 dilution); γ-tubulin (Santa Cruz Biotechnology, Cat No. sc-17787, 1:500 dilution); and HA (Santa Cruz Biotechnology, Cat No. sc-7392, 1:500 dilution); and HBx (Millipore, Cat No. MAB8419, 1:500 dilution), followed by their subsequent incubation with HRP-conjugated anti-mouse secondary antibody (Bio-Rad, Cat No. BR170-6516, 1:3000 dilution), anti-rabbit IgG (H + L)-HRP (Bio-Rad, Cat No. BR170-6515, 1:3000 dilution) or anti-goat IgG (H + L)-HRP (Thermo Scientific, Cat No. 31400, 1:10,000 dilution). An ECL kit (Advansta, San Jose, CA, USA, Cat No. K-12043-D20) was used to visualize protein bands using the ChemiDoc XRS imaging system (Bio-Rad).

### 4.8. Immunoprecipitation

An IP assay was performed using a Classic Magnetic IP/Co-IP kit (Thermo Fisher Scientific, Cat No. 88804) according to the manufacturer’s specifications. Briefly, 4 × 10^5^ cells per 60 mm diameter plate were transiently transfected with the indicated expression plasmids for 48 h under the indicated conditions. Whole-cell lysates were incubated with an anti-HBx antibody (Millipore, Cat No. 8419) overnight at 4 °C to allow the formation of the immune complexes. After washing, the immune complexes were harvested by incubation with protein A/G magnetic beads and the lysates were incubated for an additional hour. Beads were then collected using a magnetic stand (Pierce, Waltham, MA, USA), and the eluted antigen/antibody complexes were subjected to Western blotting using the assigned antibodies.

### 4.9. Cell Viability Analysis

For the determination of viable cells, an MTT assay was performed as previously described [57]. Briefly, cells were seeded at 1 × 10^4^ cells per well in 96-well plates and incubated under the indicated conditions. The cells were then treated with 10 µM 3-(4,5-dimethylthiazol-2-yl)-2,5-diphenyltetrazolium bromide (MTT, United States Biological, Cat No. 19265) for 4 h at 37 °C. The formazan compounds derived from MTT by mitochondrial dehydrogenases of the living cells were then dissolved in DMSO (Sigma-Aldrich), and quantified by measuring absorbance at 550 nm.

### 4.10. Flow Cytometry Analysis

For the determination of intracellular HBx and p53 levels by flow cytometry analysis, cells were fixed using a BD Cytofix/Cytoperm kit (BD Biosciences, Franklin Lakes, NJ, USA, Cat No. 554714). Briefly, 1 × 10^6^ cells were resuspended in a fixation/permeabilization solution at 4 °C for 20 min. A staining buffer containing 0.1% sodium azide and 2% FBS was added to the cell suspension, followed by incubation at room temperature for 10 min. The cells were reacted with anti-HBx (Santa Cruz Biotechnology, Cat No. sc-57760, 1:20 dilution) and anti-p53 (Cell Signaling, Danvers, MA, USA, Cat No. 2527S, 1:50 dilution) on ice for 30 min and then with anti-mouse IgG–FITC (Sigma-Aldrich, Cat No. F0257, 1:100 dilution) and anti-rabbit IgG–FSD 647 (Invitrogen, Cat No. A21144, 1:100) for an additional 30 min in the dark. Cells were washed twice with BD Perm/wash buffer (BD Biosciences) and then resuspended in a staining buffer or PBS. Data were obtained using a FACSAria Fusion sorter (BD Biosciences) with FACSDiva 9.0.1 software (BD Biosciences).

### 4.11. Statistical Analysis

The values indicate mean ± standard deviation from at least three independent experiments. Two-tailed Student’s *t*-test was used for all statistical analyses. A *p* value > 0.05 was considered statistically non-significant, whereas a *p* value ≤ 0.05 was considered statistically significant.

## Figures and Tables

**Figure 3 ijms-24-13354-f003:**
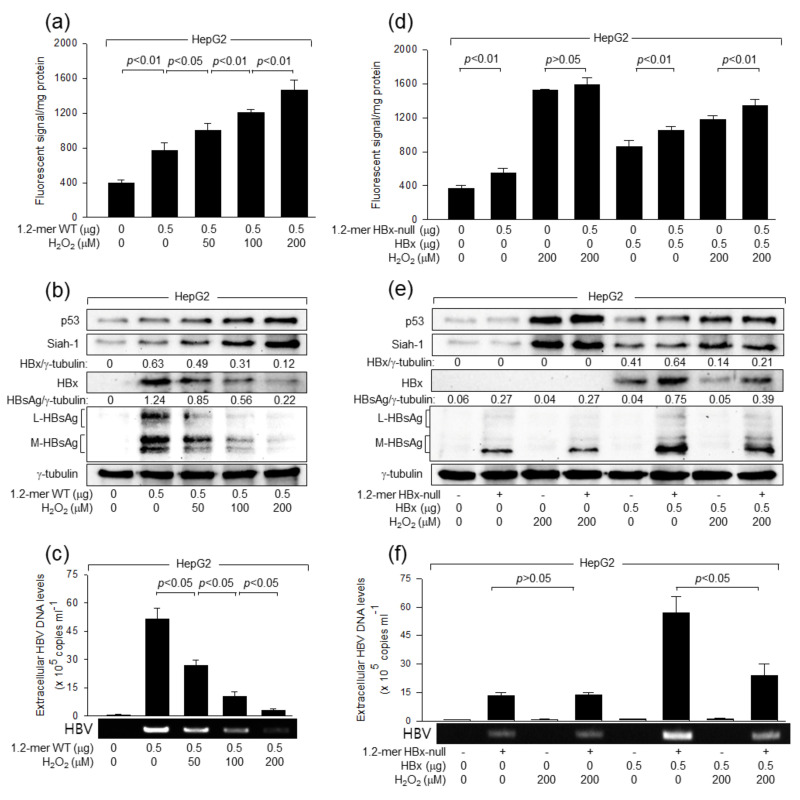
H_2_O_2_ inhibits HBV replication by lowering HBx levels. HepG2 cells were transfected with a 1.2-mer WT HBV replicon (1.2-mer WT) or its HBx-null counterpart (1.2-mer HBx-null), along with or without an HBx expression plasmid for 24 h, and then treated with the indicated concentrations of H_2_O_2_ for an additional 24 h. (**a**,**d**) Cells were subjected to ROS detection assay as described in Figure 1c. (**b**,**e**) Levels of the indicated proteins were determined via Western blotting. Levels of HBx and HBsAg were quantified as described in Figure 1a. (**c**,**f**) Levels of extracellular HBV DNA were determined by both conventional PCR and qPCR (*n* = 3).

**Figure 4 ijms-24-13354-f004:**
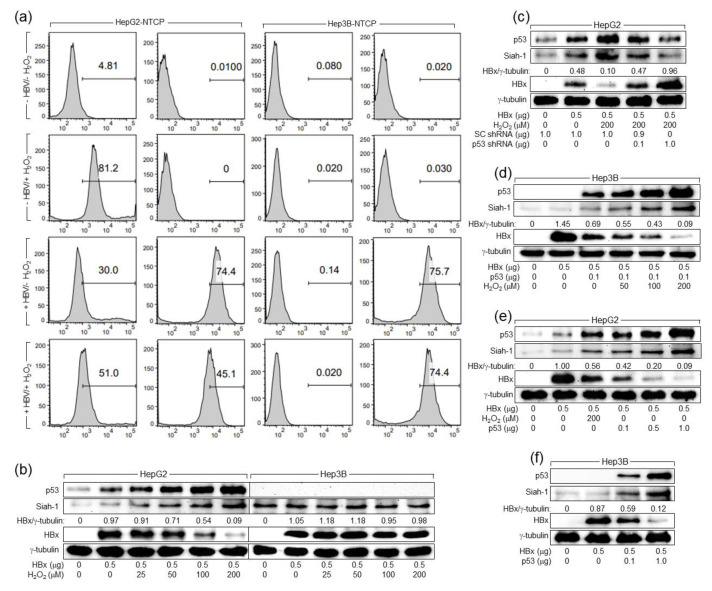
H_2_O_2_ downregulates HBx levels by elevating p53 levels in human hepatoma cells. (**a**) Flow cytometric histograms of HBV-infected HepG2-NTCP and Hep3B-NTCP cells. Cells were infected with HBV in the presence and absence of H_2_O_2_ as described in Figure 1 and then analyzed by flow cytometry to determine the proportion of cells expressing p53 and HBx. (**b**–**f**) HepG2 and Hep3B cells were transiently transfected with the indicated amounts of HBx expression plasmid along with scrambled (SC) shRNA, p53 shRNA, or p53 expression plasmid for 24 h and treated with H_2_O_2_ at the indicated concentration for an additional 24 h, followed by Western blotting. Levels of HBx were quantified as described in Figure 1a.

**Figure 5 ijms-24-13354-f005:**
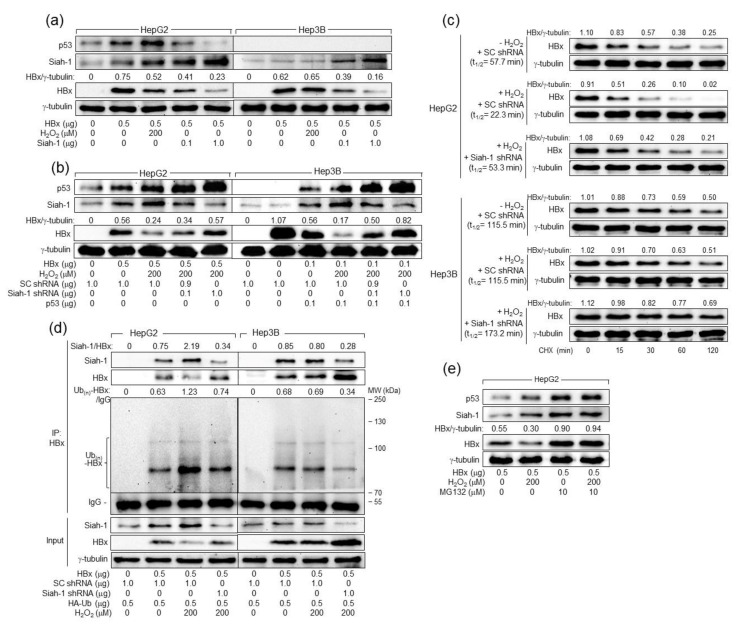
H_2_O_2_ lowers HBx levels by inducing Siah-1-mediated ubiquitination and proteasomal degradation in a p53-dependent fashion. (**a**,**b**) HepG2 and Hep3B cells were transfected with the indicated amounts of HBx expression plasmid along with Siah-1, SC shRNA, Siah-1 shRNA, or p53 expression plasmid for 24 h and treated with H_2_O_2_ at the indicated concentration for an additional 24 h, followed by Western blotting. (**c**) HepG2 and Hep3B cells prepared as in (**a**) were treated with 50 μM cycloheximide (CHX) for the indicated period before harvesting, followed by Western blotting. The levels of HBx and γ-tubulin were quantified, as described in Figure 1a, to determine the half-life (t_1/2_) of HBx. (**d**) HepG2 and Hep3B cells were transfected with the indicated plasmids for 24 h and treated with H_2_O_2_ for an additional 24 h as in (**a**). The HA-Ub expression plasmid was included in the transfection mixtures. Total HBx protein in cell lysates was immunoprecipitated with an anti-HBx antibody and subjected to Western blotting using anti-Siah-1, anti-HBx, and anti-HA antibodies to detect Siah-1, HBx, and HA-Ub-complexed HBx, respectively. The input shows the levels of the indicated proteins in the cell lysates. (**e**) HepG2 cells prepared as above were treated with 10 μM MG132 for 4 h before harvesting, followed by Western blotting.

**Figure 6 ijms-24-13354-f006:**
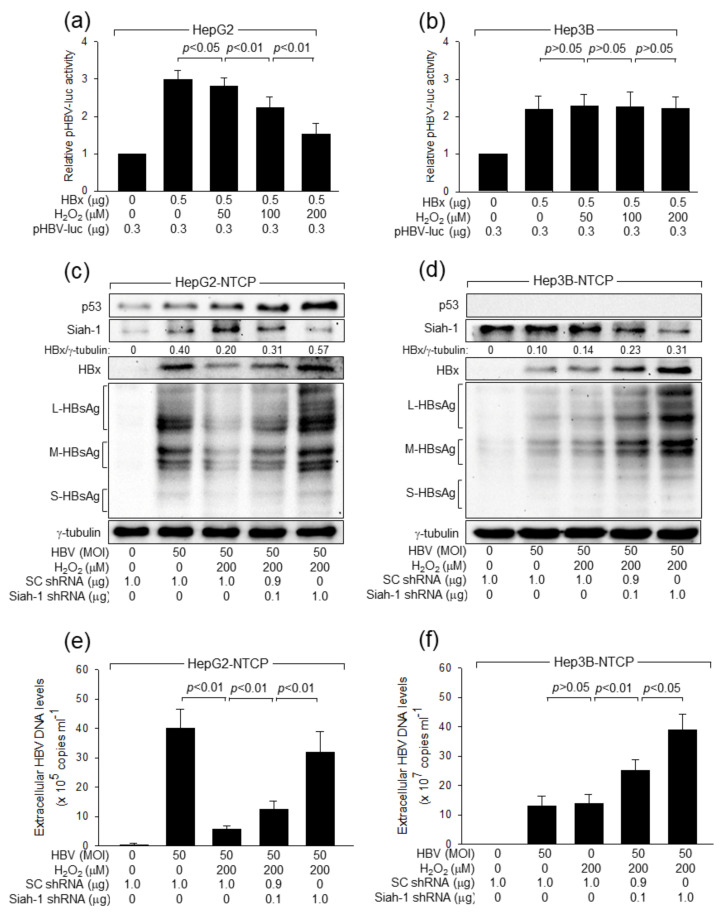
H_2_O_2_ inhibits HBV replication by inducing Siah-1-mediated proteasomal degradation of HBx in a p53-dependent fashion. (**a**,**b**) HepG2 and Hep3B cells were transiently transfected with the indicated amounts of HBx expression plasmid and pHBV-luc, which contains the HBV core promoter/enhancer, for 24 h and treated with H_2_O_2_ for an additional 24 h, followed by luciferase assay. The values show the relative luciferase activity compared to the basal level of the control (*n* = 4). (**c**,**d**) HepG2-NTCP and Hep3B-NTCP cells were transfected with SC shRNA and Siah-1 shRNA plasmids for 24 h and then either mock-infected or infected with HBV for an additional 4 days. Cells were treated with H_2_O_2_ at the indicated concentration for 24 h before harvesting, followed by Western blotting. Levels of HBx were quantified as described in Figure 1a. (**e**,**f**) The levels of HBV particles released from the cells prepared in (**c**,**d**) were determined by qPCR (*n* = 4).

## Data Availability

The data presented in this study are available from the corresponding author upon reasonable request.

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
