# Peer review of "Hydrogen Peroxide Inhibits Hepatitis B Virus Replication by Downregulating HBx Levels via Siah-1-Mediated Proteasomal Degradation in Human Hepatoma Cells"

_ijms, 2023, doi:10.3390/ijms241713354_

Round 1
Reviewer 1 Report
The submitted manuscript report by Yoon et al., proposes that H2O2 interferes with HBV replication in hepatoma cell lines in a P-53 dependent manner. The authors have proved through western blotting and other techniques the over-expression and downregulation of genes to come to the conclusion. Though rigorous work has been done for pathway analysis, there are some critical questions still needed to be addressed:
1. Fig. 1g, although defined to be "dramatic" change in ROS levels, the data doesn't reflect that. Authors need to re-analyze this one and be careful in their choice of words for explanation.
2. NAC is a well-known inhibitor of ROS, but that does not help authors to infer that its effect is ROS-specific and cannot be on other stress pathways. Moreover authors must include the role of NAC on other pathways.
3. Apart from ROS, what are the other indicators of stress response in the virus replication levels? What are LC3 levels and autophagy in the same?
4. 510-513, again authors are taking huge leap in coming to conclusion about developing antivirals targeting ROS. If so, they need to perform cell specific replication studies and not western blots that provide an overall gene level expression of all cells infected. What are the percentage of cell infected?
The manuscript needs to be sharp and focused, stressing on key findings avoiding unnecessary lengthening of their conclusions.
Author Response
1. Fig. 1g, although defined to be "dramatic" change in ROS levels, the data doesn't reflect that. Authors need to re-analyze this one and be careful in their choice of words for explanation.
Answer: We deleted some exaggerated expressions such as “dramatic” in lines 115, 126 and 142.
2. NAC is a well-known inhibitor of ROS, but that does not help authors to infer that its effect is ROS-specific and cannot be on other stress pathways. Moreover authors must include the role of NAC on other pathways.
Answer: We added new sentences in the discussion section (lines 447-457) as follows: While the present study highlights NAC as a representative antioxidant that inhibits HBV replication, it is important to consider that NAC's actions may not be solely limited to its antioxidant properties. NAC, aside from being a precursor for the antioxidant glutathione, is recognized for its interactions with various components of cell signaling pathways [53].
3. Apart from ROS, what are the other indicators of stress response in the virus replication levels? What are LC3 levels and autophagy in the same?
We added new sentences in the discussion section (lines 451-461) as follows: Besides ROS, a range of stress responses like ER stress, heat shock response, DNA damage response, apoptosis, and autophagy can all have an influence on virus replication levels. In the context of the present study, the negative effect of H2O2 on HBV replication might not result from its cytotoxicity, as demonstrated by the MTT assay data (Figure 1d).
4. 510-513, again authors are taking huge leap in coming to conclusion about developing antivirals targeting ROS. If so, they need to perform cell specific replication studies and not western blots that provide an overall gene level expression of all cells infected. What are the percentage of cell infected?
Answer: We corrected the conclusion by focusing on key findings as follows: In conclusion, the present study provides insights into the mechanism underlying the regulation of HBV replication under oxidative stresses in patients (lines 512-520). In addition, the abstract was also corrected accordingly (lines 22-25).
Comments on the Quality of English Language
The manuscript needs to be sharp and focused, stressing on key findings avoiding unnecessary lengthening of their conclusions.
Answer: The unnecessary descriptions in the discussion section were deleted based on the reviewer’s comments.
Reviewer 2 Report
In their manuscript, Hyunyoung Yoon and coauthors present the data obtained in a series of experiments aimed at the understanding the mechanism underlying the suppression of HBV replication caused by elevated levels of reactive oxygen species. Authors clearly demonstrated that H2O2 upregulates p53 to downregulate HBx during HBV replication in human hepatoma cells. Further, authors demonstrated in series of experiments in vitro, that H2O2 downregulates HBx levels via p53-dependent upregulation of ubiquitin ligase Siah-1 levels and, subsequently, via Siah-1-mediated proteasomal degradation of HBx. Finally, authors demonstrated that H2O2-induced HBx degradation resulted in decrease in HBV replication due to the reduced HBx-mediated activation of the HBV core promoter in HepG2 cells. Thus, this study provides insights into the mechanism of H2O2-induced suppression of HBV replication. The study is done well and data are presented in a clear way. I do not have any specific comments on the manuscript and assume that it can be published in a present form.
Author Response
Thanks for your generous review on our manuscript.
Reviewer 3 Report
The research article by Yoon et al., entitled “Hydrogen Peroxide Inhibits Hepatitis B Virus Replication by Downregulating HBx Levels via Siah-1-Mediated Proteasomal Degradation in Human Hepatoma Cells, investigated the role of reactive oxygen species (ROS) mainly hydrogen peroxide (H2O2), in the activation of Siah-1 expression via activation of p53 in HBV infected hepatoma cell lines and concluded that H2O2, downregulates HBx levels via Siah-1 mediated proteasomal degradation to inhibit HBV replication. The article has some interesting observations and findings based on the various experimental data shown which may be useful in understanding the HBV infection in detail and in development of antiviral against HBV. However, I have some major concerns about the article.
Major Concerns
1) Several published studies demonstrated a completely opposite role of H2O2 in the regulation of HBV replication and they concluded that, H2O2 promotes HBV replication in cell culture systems, whereas removal of ROS by treatment with NAC results in inhibition of HBV replication. Some of these studies has been mentioned and discussed by the authors in their conclusion section (page 12, line number 420-433). The authors believe that the discrepancy on this issue is due to differences in experimental conditions such as HBV genotype, host cell, and different HBV infection system. This is a very critical point and a pinpointed answer about the role of H2O2 in the HBV replication is needed. In the study authors claimed three important findings.
A) H2O2 upregulates p53 levels.
B) p53 upregulates the expression of Siah-1.
C)Siah-1 mediate proteasomal degradation of HBx to inhibit HBV replication.
In this situation, if it possible for authors, I would like to suggest following.
Take the liver biopsies of HBV infected patients and divide the cases into two groups, low replicating and high replicating based on the HBV DNA level of the case; like HBV DNA level of more than 103 can be considered as high replicating group and less than 103 can be considered as low replicating group. In the liver biopsies of these groups, see the expression level of H2O2, p53, Siah-1 and HBx. p53, Siah-1 and HBx expression levels can be checked by either immunohistochemistry or qPCR. In my opinion, data from HBV infected patient will help to resolve the discrepancies and will give a clear conclusion.
2) In the figure-2, authors have shown the experimental data in the HepG2-NTCP cells. Please perform the same experiments in the Hep3B-NTCP cells by ectopic expression of p53 and show the data of both cell lines together.
3) In the figure 3E lane 3 and 4, the protein band for p53 and Siah-1 looks, similar, although these are two different conditions, one in the present of 1.2mer HBx-null and one in the absence of 12. Mer HBx-null. Please explain and clarify these findings.
4) The Important observation from the study is about the use of HBV culture system. In most of the HBV related studies, researchers have been using HepG2-NTCP/ Hep3B-NTCP cells to study HBV replication and its biology. HBV infection elevates the intracellular ROS levels. According to the results and conclusion of this study, increased H2O2 level inhibits the HBV replication through the activation of p53. HepG2 cells express p53 however, transcripts of the p53 gene are undetectable in Hep 3B cells. My specific questions are, do the authors think, these two-culture system (HepG2-NTCP and Hep3B-NTCP) are comparable to each other in term of studying the HBV replication and biology? Authors should discuss this in detail in the discussion section.
5) The overall representation of data and figures are complex, which may not be easy to understand for the researchers who are not working in the field of HBV. The authors should reconsider a simpler way of presentation.
6) In the conclusion authors mentioned “In conclusion, the present study not only provides insights into the mechanism underlying the regulation of HBV replication under oxidative stresses in patients but also lays the groundwork for the development of ROS-inducing agents as potential anti-viral drugs against HBV infection” (page-13 line no 507). Authors should avoid the word patients, because study was performed in the culture system not in the HBV infected patients.
Some of the sentences need proper English grammar correction and rephrasing.
Author Response
1) As the reviewer commented, conducting an in vivo study using liver biopsies and sera from HBV-positive patients would indeed provide valuable confirmation of the data obtained in the current study. This approach holds the potential to offer a more comprehensive understanding of the effect of ROS on HBV replication in a clinical context. However, it's important to acknowledge that such an in vivo study is complex and would require more extensive efforts compared to the current research. Indeed, an in vivo study involving patient samples introduces various challenges. Factors such as HBV genotypes, patient genetic backgrounds, and ethical considerations need careful consideration. The potential variation among patients could impact the study's outcomes and conclusions. While an in vivo study is a logical progression, it's crucial to recognize the complexities and potential limitations associated with it. An extensive and well-designed study, perhaps building on the foundation of the current research, would be needed to effectively address these complexities. This could potentially be a future avenue of investigation, offering a more complete perspective on the interaction between ROS and HBV replication in a clinical context.
2) As the reviewer suggested, the same experiments were performed in Hep3B-NTCP cells by ectopic expression of p53 as shown in Figure 2d to f.
3) Indeed, the transfection of 1.2-mer HBx-null had limited influence on p53 and Siah-1 levels in HepG2 cells, irrespective of H2O2 treatment. This observation serves to validate the role of HBx as an activator of p53. The results indicate that the absence of HBx does not substantially alter the levels of p53 and Siah-1, and this remains unaffected by the presence of H2O2. This reinforces the notion that HBx plays a pivotal role in stimulating p53 activity, and its absence does not significantly impact p53 and Siah-1 levels. This outcome underscores the contribution of HBx in modulating p53 activation in response to oxidative stress.
4) Please refer to the new descriptions in the Discussion section (lines 474 to 485) as follows: In this study, two human liver cancer cell lines, HepG2 and Hep3B, were employed to investigate the potential role of p53 in the regulation of HBV replication by H2O2, because HepG2 cells but not Hep3B cells express a functional form of p53, while sharing several well-characterized characteristics [36,54]. H2O2 inhibited HBV replication in HepG2-NTCP cells, in which H2O2 upregulated p53 levels (Figure 1a and b), but not in Hep3B-NTCP cells (Figure 1e and f), suggesting a possible role of p53 in the regulation of HBV replication. However, the two cell lines exhibit several other differences, such as ethnic origins, distinct chromosome abnormalities, HBV DNA integration, and tumorigenicity [36,54], which may affect the effect of H2O2 in these two cell lines. This question can be answered by examining the effect of H2O2 in Hep3B cells after p53 complementation and in HepG2 cells after p53 knockdown. Indeed, ectopic p53 expression successfully restored the potential of H2O2 to inhibit HBV replication in Hep3B-NTCP cells, in which H2O2 upregulated exogenous p53 levels (Figure 1i and j). Moreover, p53 knockdown almost abolished the potential of H2O2 to downregulate HBx levels in HepG2 cells (Figure 4c).
5) We hesitate to provide a definite response regarding the reviewer's comment on the complexity of the data presentation in the manuscript. We understand that different reviewers might have varied perspectives on data presentation based on their individual preferences and experiences. However, given our extensive publication history in the field of HBV and HCV, where similar data presentation styles have been utilized successfully, we want to assert our approach's consistency established in the previous publications.
6) We provided a simple and concise conclusion by focusing on the primary findings of the present study as follows: “In conclusion, the present study provides insights into the mechanism underlying the regulation of HBV replication under oxidative stresses in patients.” (lines 535-537)
7) Comments on the Quality of English Language
Some sentences were corrected and rephrased.
Round 2
Reviewer 1 Report
The authors have made considerable improvement to their manuscript and addressed all the questions asked. The manuscript looks enhanced. I further have no issues.
Author Response
Thanks for your generous consideration on our manuscript.
Reviewer 3 Report
Authors successfully answered all the issues raised, I have no further questions.